# Assessment of Health Beliefs and Motivations for Cervical Cancer Screening

**DOI:** 10.3390/healthcare13070804

**Published:** 2025-04-03

**Authors:** Gülsüm Hatice Yüksel, Ayşe Nilüfer Özaydın

**Affiliations:** 1Health Sciences Institute, Marmara University, 34865 İstanbul, Türkiye; 2Department of Public Health, School of Medicine, Internal Medical Sciences, Marmara University, 34854 İstanbul, Türkiye; aozaydin@marmara.edu.tr

**Keywords:** primary care, prevention, health belief model, screening motivation, sociodemographic factors, STD risk perception

## Abstract

**Background and Aim:** This study evaluated Turkish women’s health beliefs and motivations for participating in cervical cancer screening using the Health Belief Model Scale for Cervical Cancer and Pap Smear Test (HBMS-CCPST). Although the HPV vaccine has proven effective, Turkey has not integrated it into the national health program, but it provides free HPV-DNA screenings for women at family health centers (FHCs). This study evaluated Turkish women’s health beliefs and motivations for participating in cervical cancer screening using the Health Belief Model Scale for Cervical Cancer and Pap Smear Test (HBMS-CCPST) with the aim of understanding and enhancing screening uptake. **Methods:** This cross-sectional study was conducted at the family health centers in the Kartal and Pendik districts of Istanbul and involved women aged 30–65 years. The participants were randomly selected from the FHCs’ 2023 lists. Data were collected through face-to-face interviews, using a structured questionnaire based on the HBMS-CCPST. **Results:** A randomly selected sample of 422 women from 8003 registered FHCs was approached to participate; 354 agreed, 25 declined, and 43 were excluded because they did not meet the inclusion criteria. The mean age was 44.58 years (range: 30–64), and 51% had previously undergone a smear test. Correlations were found between age and the total HBMS-CCPST score (r = −0.207, *p* = 0.001), perceived benefits (r = −0.106, *p* = 0.046), health motivation (r = −0.195, *p* = 0.001), and perceived barriers (r = 0.132, *p* = 0.013). Social security status influenced the HBMS-CCPST scores (*p* = 0.019), health motivation (*p* = 0.011), and perceived barriers (*p* = 0.002). Employment status affected the total score (*p* = 0.001), health motivation (*p* = 0.001), and perceived barriers (*p* = 0.001), with employed women showing higher health motivation and lower perceived barriers. Alcohol use and physical activity levels affected the total score (*p* = 0.001 and *p* = 0.007, respectively) and health motivation (*p* = 0.011 and *p* = 0.001, respectively). STD risk perception affected the HBMS-CCPST scores (*p* = 0.001, *p* = 0.001) and perceived barriers (*p* = 0.001 and *p* = 0.001, respectively). **Conclusions:** Sociodemographic factors and daily habits significantly influenced the participants’ health beliefs and motivations for cervical cancer screening.

## 1. Introduction

The global strategy to accelerate the elimination of cervical cancer as a public health problem and its associated goals and targets was defined for the period 2020–2030 (73rd World Health Assembly). Cervical cancer remains a significant global health concern, particularly in developing countries, where it is the second most common cancer among women. According to the latest GLOBOCAN 2022–2024 data, there are approximately 604,000 new cases and 342,000 deaths annually worldwide, highlighting the urgent need for effective prevention and screening strategies. In developing countries, it is the second most common cancer, with over 400,000 cases and approximately 230,000 deaths per year, accounting for more than 80% of the global burden. The higher mortality rate in these regions is due to the minimal investment in cancer prevention programs, which is approximately 5% of that in developed countries. Cervical cancer is a critical health concern among women in Turkey. In 2018, the Turkish Ministry of Health reported an incidence rate of 4.4 per 100,000 women, with approximately 1500 new cases and 650 deaths annually. Over 30% of Turkish women diagnosed with cervical cancer are in the advanced stages, indicating inadequate screening and prevention efforts [1].

A strong link has been established between cervical cancer and human papillomavirus (HPV), particularly serotypes 16 and 18, which account for nearly 70% of all HPV infections worldwide. HPV infection causes changes in the cervical epithelium, particularly during puberty. The World Health Organization recommends vaccination at ages 9 to 14 as a highly effective way to prevent HPV infection, cervical cancer, and other HPV-related cancers [2].

Primary prevention strategies, such as HPV vaccination, along with secondary measures, including the screening and treatment of precancerous conditions, can effectively prevent the majority of cervical cancer cases. Cervical cancer, when identified early, is among the most treatable cancers, with successful outcomes through effective management. The advanced stages of the disease can be managed with suitable treatment and palliative care options. By adopting an all-encompassing strategy to prevent, detect, and treat cervical cancer, it is feasible to eradicate it as a major public health issue within a single generation [3].

Cervical cancer screening programs, notably Pap smears, are pivotal in the early detection of precancerous lesions. These programs have successfully reduced the incidence and mortality rates of cervical cancer by approximately 70% over three years in developed nations. The protracted development of cervical cancer from detectable precancerous lesions places women aged 30–40 at a heightened risk. In line with this, Turkish health guidelines advocate for universal screening, recommending a combined Pap smear and HPV test every five years for women aged 30–65, with an option to extend this interval [4,5,6].

The decision to participate in these preventive or detection programs is significantly influenced by factors outlined in the Health Belief Model Scale for Cervical Cancer and Pap Smear Test (HBMS-CCPST). This model evaluates perceived susceptibility to the disease, perceived severity, perceived benefits of screening, and perceived barriers to screening methods. The HBMS-CCPST has been extensively validated and applied across different cultural contexts [7], although its specific application in Turkey has been limited [1,8].

Screening programs significantly reduce cervical cancer incidence and mortality rates when administered effectively, demonstrating the critical role of health policies in cancer prevention.

This study aimed to delve deeper into the health beliefs of participants regarding cervical cancer and the HPV-DNA test. It explored how these beliefs correlate with their knowledge and understanding of cervical cancer and Pap smear tests, along with their sociodemographic characteristics. By focusing on these aspects, this study seeks to provide insightful data that could inform more effective health communication strategies and screening programs.

## 2. Materials and Methods

### 2.1. Study Design

This cross-sectional study was conducted at the family health centers in the Kartal and Pendik districts of Istanbul. The study was conducted on women who were registered and regularly followed up at their health centers. All the participants were randomly selected, informed, and invited to participate in the study. Ethical considerations were strictly followed during the data collection process to ensure the privacy and confidentiality of the participants.

### 2.2. Ethical Considerations

Permission for research was sought from the Marmara University Institute of Health Sciences on 22 September 2023, and approval was granted on 12 October 2023 (decision No. 2023/35). Ethical approval for the study was obtained from the Ethics Committee on 6 October 2023 (protocol code 09.2023.1266). Approval from the Istanbul Provincial Health Directorate was obtained on 28 November 2023 (decision No. 2023/19).

### 2.3. Sample Population

This study focused on the districts of Pendik and Kartal in Istanbul. In August 2023, a total of 3588 women aged 30–65 were registered in five family medicine units at the Kartal Family Health Center, while 4415 women in the same age group were registered in five family medicine units in Pendik (Table 1 and Table 2).

Each family medicine unit was accepted as a subgroup, and a stratified sample selection was made from women registered in each unit according to their age groups (30–34, 35–39, 40–44, 45–49, 50–54, 55–59, 60–64 years). The sample calculated from each subgroup was selected in proportion to the population using a simple random method.

In accordance with the literature, the sample size was calculated to be 176 for each group, and the total sample size of 352 was increased by 20% by considering an additional 20% rejection probability. The sample size was calculated as 211 to obtain baseline data from the intervention and control groups (422 in total).

### 2.4. Eligibility Criteria

This study targeted women who met the following inclusion criteria.

Age of 30 to 65 years.Registered and regularly followed up at either of the two family health centers located in Kartal and Pendik, Istanbul.Able to provide informed consent and willing to participate in the study.Fluent in Turkish to ensure clear communication during the interviews.

The exclusion criteria were as follows.

Inability to speak Turkish.Inaccessibility or emigration.Ongoing pregnancy.Undergoing a hysterectomy or diagnosis of cervical neoplasia.Illiteracy.Refusal to participate in the study.

### 2.5. Data Collection

Data were collected through face-to-face interviews using a structured questionnaire. The interviews were conducted in a private setting at the family health center to ensure the confidentiality and comfort of the participants.

The questionnaire covered aspects of the health belief model, including perceived susceptibility, perceived severity, perceived benefits, and perceived barriers related to cervical cancer and the Pap smear test. This scale, developed by Champion for breast cancer and mammography, was adapted for cervical cancer and the Pap smear test. A Turkish validity and reliability study was conducted by Güvenç, Akyüz, and Açıkel in 2010. The scale consists of 35 items and five main dimensions: sensitivity (3 items), seriousness (7 items), Pap smear benefits and motivation (8 items), health motivation (3 items), and Pap smear barriers (14 items). A 5-point Likert-type scale ranging from 1 to 5 (1 = strongly disagree, 2 = disagree, 3 = undecided, 4 = agree, 5 = strongly agree) was used to evaluate the scale. Each dimension of the scale was evaluated separately. Higher scores indicate higher sensitivity, importance, and motivation; higher benefits are perceived for the perception of benefits, and higher barriers are perceived for the perception of barriers [9].

### 2.6. Statistical Analysis

The collected data were analyzed using statistical software (SPSS for Mac ver. 29). Descriptive statistics were used to summarize the demographic characteristics of the participants. Associations between sociodemographic variables and health beliefs regarding cervical cancer and the Pap smear test were analyzed using the chi-squared test. Spearman’s rank correlation coefficient was used to examine the relationships between continuous variables. Statistical significance was set at *p* = 0.05.

## 3. Results

A total of 422 women registered at family health centers in Kartal and Pendik were approached to participate in the study. Of these, 354 women met the inclusion criteria and agreed to participate, while 25 refused to participate; 43 were excluded because they did not meet the inclusion criteria. Thus, the overall participation rate, considering those eligible and approached, was 84% (Figure 1).

### 3.1. Sociodemographic Characteristics of the Participants

The mean age was 44.58 years (SD, range: 30–64 years). The majority of the participants were covered by the Social Security Institution (SSI), with 86.4% having such coverage, reflecting a high rate of insured individuals in the study population. Other demographic details, including education, employment, income levels, marital status, childbearing status, health status, and smoking habits, are detailed in Table 3.

### 3.2. The Health Belief Model Scale for Cervical Cancer and Pap Smear Test Results

Age was inversely correlated with several components of the health belief model. As age increased, there was a decrease in perceived benefits (Spearman’s rho = −0.106, *p* = 0.046) and health motivation (Spearman’s rho = −0.195, *p* = 0.001) and an increase in perceived barriers (Spearman’s rho = 0.132, *p* = 0.013), indicating that older participants felt less motivated and more hindered regarding cervical cancer screening.

Marital status also impacted health motivation as individuals who were never married exhibited higher motivation than those who were married or divorced (*p* = 0.007).

Having children also affected participants’ health beliefs. Those with children had significantly different HBMS-CCPST scores (*p* = 0.022) and higher health motivation (*p* = 0.001) than those without children. This suggests that parental responsibilities might increase awareness and motivation regarding health prevention measures.

Social security status significantly influenced the Health Belief Model Scale for Cervical Cancer and the Pap Smear Test (HBMS-CCPST) scores. Notably, participants without social security experienced higher perceived barriers (*p* = 0.002) than those with social security, including Social Security Institution (SSI), private insurance (PI), and other types. The HBMS-CCPST scores (*p* = 0.019) and health motivation (*p* = 0.011) also varied significantly according to social security status.

Employment status was another determinant factor, as currently employed participants reported higher health motivation and faced fewer perceived barriers than those who were unemployed or previously employed (*p* = 0.001 for the total score, health motivation, and perceived barriers).

Table 4 presents a detailed analysis of the HBMS-CCPST results across various sociodemographic variables, including age, social security status, employment status, marital status, and number of children.

Regarding health status, there were significant differences in perceived susceptibility (*p* = 0.027) and perceived barriers (*p* = 0.046) among the participants with different health statuses. Participants with excellent or very good health had higher health motivation and lower perceived barriers than those with poor health. Chronic disease status showed significant differences in the total score (*p* = 0.014), with the participants with chronic diseases showing different scores compared to those without chronic diseases. For alcohol use status, significant differences were found in the total score (*p* = 0.001), health motivation (*p* = 0.011), and perceived barriers (*p* = 0.014). The participants who consumed alcohol had higher total scores and health motivation, but lower perceived barriers than those who did not consume alcohol. Physical activity showed significant differences in the total score (*p* = 0.007), health motivation (*p* = 0.001), and perceived benefits (*p* = 0.077). The participants who exercised vigorously three days a week had higher total scores and health motivation than those who did not exercise or exercised lightly. Table 5 presents the analysis of HBMS results based on various health statuses and daily habit variables, including health status, chronic disease status, smoking status, alcohol use status, and physical activity.

Regarding the contraceptive use status, no significant differences were found in the total score, HBMS-CCPST, or any sub-scores (*p*-values ranging from 0.268 to 0.965). Both groups (users and non-users) had similar perceptions of susceptibility, severity, benefits, health motivation, and barriers. Regarding STD risk perception, significant differences were observed in the HBMS-CCPST scores (*p* = 0.001), perceived susceptibility (*p* = 0.022), perceived severity (*p* = 0.049), and perceived barriers (*p* = 0.001). The participants who perceived themselves to be at risk had higher HBMS-CCPST scores and perceived barriers than those who did not perceive any risk or had no idea about their risk. The need for screening showed significant differences in the total score (*p* = 0.001) and perceived susceptibility (*p* = 0.001). No significant differences were found in the HBMS-CCPST or other sub-scores (*p*-values ranging from 0.140 to 0.896). Perceived difficulty in accessing information about recommended screening procedures significantly affected the total score (*p* = 0.001), HBMS-CCPST (*p* = 0.002), health motivation (*p* = 0.032), and perceived barriers (*p* = 0.001). The participants who found it very difficult or difficult to find information had lower total scores and health motivation and higher perceived barriers than those who found it easy or had no idea about it. Table 6 presents the analysis of HBMS results based on various sexual health variables, including contraceptive use status, STD risk perception, need for screening, and difficulty in finding information.

## 4. Discussion

This study effectively explored the health beliefs and motivations of Turkish women regarding cervical cancer screening using the HBMS-CCPS test. One of the key strengths of this study is its detailed examination of the interplay between sociodemographic factors, including age, employment status, social security coverage, and health beliefs and behaviors. The findings revealed significant influences of these factors on the willingness to participate in cervical cancer screening, highlighting potential targets for public health interventions. Conducted within the cultural and social context of Istanbul, this study provides insights into the specific barriers and facilitators affecting cervical cancer screening. By providing a comprehensive view of how health status and lifestyle factors, such as alcohol use and physical activity, impact health motivations, this study underscores the need for tailored health promotion strategies that could effectively increase screening rates among diverse populations in Turkey. These aspects make this study a significant contribution to the ongoing efforts to enhance cervical cancer screening and prevention strategies in the region.

The results revealed significant correlations between age and various components of the HBMS-CCPST. Specifically, older women had lower total scores and perceived benefits, but higher perceived barriers. This is consistent with the findings of Sandstorm et al., who reported that older women tend to have lower perceived benefits and higher perceived barriers to cervical cancer screening. These findings highlight the need for targeted interventions to address age-specific barriers and enhance the perceived benefits of screening among older women [10].

Perceived benefits and susceptibility emerged as the strongest predictors of Pap smear test participation. Women who acknowledged the advantages of early detection and treatment of cervical cancer were more likely to engage in screening. This aligns with previous research indicating that perceived benefits play a key role in health behavior decisions [11]. Ford et al. also reported that US Hispanic women who perceived significant benefits of cervical cancer screening, such as early cancer detection and reassurance of health, were more likely to undergo Pap smears [12]. Perceived susceptibility was another significant predictor, indicating that women who felt at risk of developing cervical cancer were more inclined to be screened. This finding is consistent with those of Chisale et al. and Agarwal et al., who reported that higher perceived susceptibility is associated with increased cervical cancer screening rates [13,14]. However, our study found that many women had low perceptions of susceptibility, often due to misconceptions about risks such as being young or not using hormonal contraceptives.

Social security status showed significant differences in the HBMS-CCPST scores, health motivation, and perceived barriers. The participants without social security had higher perceived barriers than those with SSI, PI, or other types of social security. This aligns with the study by Kuroki et al., which found that a lack of health insurance is a significant barrier to cervical cancer screening [15]. Press et al., too, emphasized that enhancing access to affordable healthcare and providing education on the importance of screening could mitigate these barriers [16]. Employment status significantly affected the total score, health motivation, and perceived barriers. Women who were currently working exhibited higher health motivation and lower perceived barriers than those who had never worked or used to work. Similar findings were reported by Aldohaian et al. [17], who suggested that employed women may have better access to healthcare services and information, leading to higher health motivation.

There was a significant association between marital status and health motivation, with never-married women reporting higher health motivation scores compared to their married or divorced counterparts. This finding contrasts with those of previous studies, such as by Momeni et al. and Baharum et al., who reported that married women had higher health motivation for cervical cancer screening [18,19]. Further research is needed to explore the underlying reasons for differences in health motivation according to marital status. The number of children significantly affected the total score and health motivation, with participants having children showing different scores from those without children. This result supports the findings of Siseho et al., who found that women with children are more likely to participate in cervical cancer screening programs because of their increased health consciousness [20].

Health and chronic disease status were significant predictors of perceived susceptibility and barriers. Women with poorer health status and chronic diseases had higher perceived susceptibility and perceived barriers. This is consistent with the findings of Ma et al., who reported that perceived susceptibility was higher among individuals with poorer health status [21]. Smoking status and alcohol use also showed significant differences among the various components of the HBMS-CCPST. Women who did not use alcohol had lower health motivation and higher perceived barriers. On the other hand, Aldohaian et al. reported that individuals who did not smoke or drink alcohol were more likely to engage in health-promoting behaviors, including cervical cancer screening. [17].

Physical activity levels were associated with significant differences in the total scores and health motivation. Women who engaged in regular physical activity had higher health motivation scores, consistent with the findings of Gorina et al., who indicated that physical activity was positively associated with health motivation and health-promoting behaviors [22]. Perceived barriers negatively affected the likelihood of undergoing a Pap smear. Common barriers included fear of test results, lack of awareness, and perceived low susceptibility. These barriers are consistent with the findings of Ahmed et al., who highlighted that access to healthcare and fear of cancer diagnosis are significant obstacles to cervical cancer screening [23].

Knowledge of cervical cancer and the Pap smear is another critical factor. Women with higher knowledge scores are more likely to undergo screening. This is in line with multiple studies, which reported that knowledge significantly impacts cervical cancer screening behaviors [13,14,18,20,21,23]. Educational interventions that target knowledge improvement can enhance screening rates. STD risk perception and perceived difficulty in accessing information about recommended screening procedures are also significant predictors of the HBMS-CCPST scores. Women with higher STD risk perception and those who found it difficult to find health information had higher perceived barriers and lower health motivation. These findings are consistent with the literature, highlighting the need for comprehensive sexual health education and improved access to reliable health information to enhance screening behaviors [13,14,18,20,21,23].

Comparatively, our findings align with global trends, where higher health motivation correlates with reduced perceived barriers, similar to studies conducted in the USA and Europe. This cross-cultural consistency underscores the universal importance of health motivation in promoting cervical cancer screenings.

This study underscores the complex interplay among sociodemographic factors, health status, and daily habits in shaping health beliefs and motivations for cervical cancer screening among Turkish women. In light of these findings, policies aimed at improving access to healthcare services and providing comprehensive sexual health education are crucial for increasing screening rates.

### Study Limitations

A significant limitation of this study is that it was conducted exclusively at family health centers in the Kartal and Pendik districts of Istanbul, and the findings may not be generalizable to women living in other districts or regions. Additionally, this study relied on self-reported data, which may introduce response bias; the participants could provide socially desirable responses or may not have accurately remembered their health behaviors and perceptions. Furthermore, the research did not consider other potential confounding factors, such as cultural influences, insurance status, employment, or access to healthcare services, which could significantly impact health beliefs and screening behaviors.

## 5. Conclusions

This study provides essential insights into the health beliefs and motivations of Turkish women aged 30–64 residing in two districts of Istanbul regarding cervical cancer and Pap smear screening, as evaluated through the HBMS-CCPS test. Key findings demonstrated that age, social security status, and employment significantly influenced perceived benefits, health motivation, and perceived barriers to cervical cancer screening. Specifically, women who were currently employed exhibited higher health motivation (mean score: 10, *p* = 0.001) and lower perceived barriers (mean score: 31, *p* = 0.001) compared to those who had never worked or used to work. Health status also played a crucial role, with those in excellent or very good health showing higher health motivation. Furthermore, perceptions of STD risk and the need for screening were strongly associated with perceived barriers. Difficulty in finding information about cervical cancer and the Pap smear significantly affected health beliefs and motivations, emphasizing the need for accessible and comprehensible health information.

The findings of this study have several important clinical implications. Healthcare providers should consider sociodemographic factors, health status, and daily habits when designing interventions that promote cervical cancer screening. Tailored educational programs that address specific barriers and enhance health motivation could significantly improve screening rates. By addressing these factors through targeted interventions and policies, healthcare professionals can enhance cervical cancer screening rates and reduce the burden of cervical cancer.

This study underscores the importance of targeted health education and intervention programs to enhance cervical cancer screening in Turkish women. Addressing perceived barriers, enhancing health motivation, and providing tailored information based on individual health status and sociodemographic characteristics could improve screening. Future research should explore longitudinal designs to establish causal relationships and consider larger and more diverse samples to enhance the generalizability of the findings. This study contributes to a better understanding of the factors influencing cervical cancer screening behaviors, which are critical for developing effective public health strategies and policies.

## Figures and Tables

**Figure 1 healthcare-13-00804-f001:**
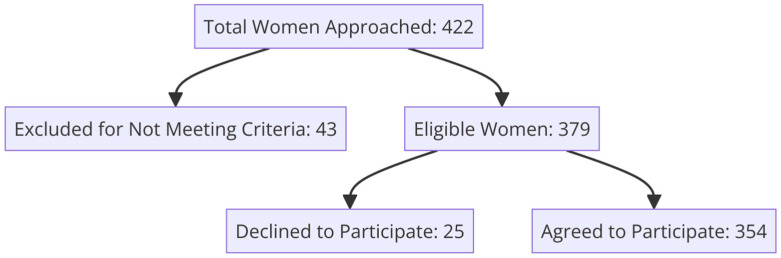
Participant selection and enrollment process flowchart.

**Table 1 healthcare-13-00804-t001:** Distribution of women registered in the family health center in Kartal according to age group and family physician unit.

Age Group	Family Physician Unit (FPU)					
	FPU No. 62	FPU No. 63	FPU No. 64	FPU No. 134	FPU No. 139	Total
30–34 years	136	127	142	169	79	653
35–39 years	118	124	149	134	61	586
40–44 years	110	112	179	106	53	560
45–49 years	95	111	156	112	49	523
50–54 years	104	109	160	110	47	530
55–59 years	87	79	124	98	35	423
60–64 years	56	56	96	80	25	313
Total	706	718	1006	809	349	3588

**Table 2 healthcare-13-00804-t002:** Distribution of women registered in the family health center in Pendik according to age group and family physician unit.

Age Group	Family Physician Unit (FPU)					
	FPU No. 94	FPU No. 95	FPU No. 96	FPU No. 97	FPU No. 98	Total
30–34 years	160	177	195	170	137	839
35–39 years	174	165	157	168	151	815
40–44 years	179	165	163	147	169	823
45–49 years	141	138	120	136	120	655
50–54 years	119	101	108	95	115	538
55–59 years	77	93	95	75	103	443
60–64 years	62	47	58	68	67	302
Total	912	886	896	859	862	4415

**Table 3 healthcare-13-00804-t003:** Sociodemographic and health characteristics of the participants.

Participants	Count (N, 354)	Frequency (%)
Social security status		
SSI coverage	306	86.4
Without health insurance	30	8.5
PI or other	18	5.1
Education level		
Literate	18	5.1
Primary School completed	116	32.8
Secondary school completed	56	15.8
High school completed	86	24.3
College and above completed	78	22.0
Employment status		
Currently working	118	33.3
Currently not working	236	66.7
Income level		
Upper-income bracket	32	9
Middle-income bracket	263	74.3
Low-income bracket	59	16.7
Marital status		
Married	294	83.1
Not married	60	16.9
Having children		
Yes	308	87
No	46	13
Health status		
Very good	19	5.4
Good	193	54.5
Fair	127	35.9
Bad	15	4.2
Smoking habits		
Non-smokers	242	68.3
Smokers	61	17.3
Occasional smokers	42	11.9
Former smokers	9	2.5
Disease presence		
With disease	116	32.8
Without disease	238	67.2
Total	354	100.0

**Table 4 healthcare-13-00804-t004:** Health Belief Model Scale results of the participants by sociodemographic variables.

Variables	Total Score	HBMS-CCPST	Perceived Susceptibility	Perceived Severity	Perceived Benefits	Health Motivation	Perceived Barriers
Age (years)							
Mean age (30–64)	7.294 ± 3.482	103.992 ± 11.645	7.771 ± 2.195	21.816 ± 5.366	31.777 ± 5.496	9.331 ± 2.354	33.297 ± 8.104
Spearman’s rho	−0.207	−0.028	−0.065	−0.024	−0.106	−0.195	0.132
*p*-value	0.001	0.600	0.224	0.647	0.046	0.000	0.013
Social security status							
None	6 (0–12)	110.367 ± 11.285	7 (3–12)	23 (9–33)	32 (22–40)	8 (6–12)	38.5 (28–58)
SSI	7 (0–17)	103.431 ± 11.418	8 (3–13)	22 (9–35)	32 (8–40)	10 (3–15)	32 (14–58)
PI	7 (4–14)	102.4 ± 16.44	8.5 (5–10)	23.5 (12–32)	32 (13–38)	9.5 (6–14)	34.5 (19–55)
Others	6.5 (2–12)	103.5 ± 9.532	6 (3–12)	24.5 (14–26)	32 (29–34)	12 (7–14)	31.5 (25–39)
*p*-value	0.544	0.019	0.443	0.585	0.951	0.011	0.002
Employment status							
Never worked	6 (0–15)	104.5 (72–130)	7.5 (3–13)	22 (9–35)	32 (8–40)	9 (3–13)	34 (14–58)
Currently working	8 (2–17)	102 (61–137)	8 (3–13)	22 (9–34)	32 (8–40)	10 (3–15)	31 (14–58)
Used to work	7 (0–15)	104 (82–129)	8 (3–12)	23 (9–33)	32 (15–40)	9 (4–15)	32 (14–51)
*p*-value	0.001	0.260	0.823	0.392	0.134	0.000	0.002
Marital status							
First and only marriage	6 (0–17)	104.269 ± 11.342	8 (3–13)	22 (9–35)	32 (8–40)	9 (3–15)	33 (14–58)
Others	6 (3–16)	106.733 ± 10.18	9 (3–12)	21 (14–32)	32 (24–40)	9 (5–12)	35 (23–54)
*p*-value	0.425	0.411	0.168	0.803	0.051	0.899	0.105
Having children							
None	7.5 (0–17)	105.325 ± 10.608	8 (3–11)	25 (9–32)	33 (15–40)	10 (6–15)	31.5 (18–45)
Yes	6 (0–17)	103.692 ± 11.661	8 (3–13)	22 (9–35)	32 (8–40)	9 (3–15)	34 (14–58)
*p*-value	0.022	0.401	0.862	0.068	0.060	0.000	0.074
Contraceptive use status							
User	7.1 ± 2.3	103.8 ± 11.4	8.2 ± 1.9	22.1 ± 4.5	31.8 ± 5.0	9.4 ± 2.2	32.7 ± 7.1
Non-user	6.9 ± 2.5	104.0 ± 12.0	8.0 ± 2.0	22.0 ± 4.6	31.5 ± 5.2	9.1 ± 2.4	33.0 ± 7.5
*p*-value	0.652	0.809	0.734	0.882	0.759	0.648	0.671
Number of marriages							
First marriage	7.0 ± 2.4	104.2 ± 11.5	8.1 ± 2.1	22.3 ± 4.7	31.7 ± 5.1	9.3 ± 2.3	32.8 ± 7.2
Subsequent marriages	6.8 ± 2.6	103.5 ± 12.1	7.8 ± 2.2	21.7 ± 4.8	31.2 ± 5.3	8.9 ± 2.5	33.3 ± 7.4
*p*-value	0.712	0.629	0.687	0.780	0.654	0.725	0.691

Kruskal–Wallis H test; one-way analysis of variance; HBMS-CCPST, Health Belief Model Scale for Cervical Cancer and Pap Smear Test.

**Table 5 healthcare-13-00804-t005:** Health Belief Model Scale results for the participants by health and daily habit variables.

Variables	Total Score	HBMS-CCPST	Perceived Susceptibility	Perceived Severity	Perceived Benefits	Health Motivation	Perceived Barriers
Health status							
Excellent/very good	8 (3–14)	102 ± 14.415	7 (4–11)	20 (10–33)	34 (16–40)	10 (6–14)	32 (15–43)
Good	6 (1–17)	102.497 ± 11.112	7 (3–12)	22 (9–35)	32 (8–40)	10 (3–15)	32 (14–55)
Fair	7 (0–16)	106.819 ± 11.868	8 (3–13)	23 (9–33)	32 (14–40)	9 (4–15)	34 (14–58)
Poor	5 (0–12)	101.8 ± 8.654	8 (3–12)	20 (14–28)	32 (20–40)	9 (6–12)	32 (28–38)
Test statistic	5.794	4.017	9.170	7.415	4.303	7.117	8.016
*p*-value	0.122	0.008	0.027	0.060	0.231	0.068	0.046
Chronic disease status							
Yes	6 (0–15)	103.698 ± 11.541	7 (3–13)	22 (9–33)	32 (12–40)	10 (4–15)	32 (14–56)
No	7 (0–17)	104.135 ± 11.717	8 (3–13)	23 (9–35)	32 (8–40)	9 (3–15)	33 (14–58)
*p*-value	0.014	0.741	0.494	0.223	0.571	0.980	0.359
Smoking status							
Non-smoker	6 (0–17)	104.195 ± 11.358	8 (3–13)	22 (9–35)	32 (9–40)	9 (3–15)	33 (14–58)
Smoker	7 (1–16)	101.902 ± 10.492	8 (3–13)	21 (9–33)	32 (8–40)	10 (3–14)	31 (14–55)
Occasional smoker	7 (0–16)	105.238 ± 13.926	7 (4–12)	23 (13–33)	32 (8–40)	10 (4–12)	34 (18–50)
Former smoker	7 (2–15)	106.667 ± 15.572	8 (3–12)	23 (12–27)	33 (14–40)	12 (6–13)	32 (23–58)
*p*-value	0.466	0.395	0.810	0.321	0.219	0.172	0.113
Alcohol use status							
Yes	9 (4–17)	104 (61–130)	9 (4–11)	23 (12–32)	32 (8–40)	10 (6–15)	30 (14–55)
No	6 (0–17)	103 (65–137)	8 (3–13)	22 (9–35)	32 (8–40)	9 (3–15)	34 (14–58)
*p*-value	0.001	0.864	0.289	0.199	0.104	0.011	0.014
Physical activity							
Never	6 (0–16)	103 (73–130)	8 (3–13)	22 (9–35)	32 (8–40)	9 (3–15)	33 (14–58)
Occasionally exercises	7 (0–17)	104 (65–137)	8 (3–13)	22 (9–33)	32 (9–40)	10 (3–15)	32 (14–58)
Exercises lightly 3 days/week	7 (1–15)	106 (61–127)	6.5 (4–12)	24 (13–32)	32 (8–40)	12 (4–15)	32 (14–45)
Exercises vigorously 3 days/week	11 (6–17)	105 (90–117)	8 (3–9)	17.5 (9–27)	35 (31–39)	11.5 (10–14)	31.5 (27–39)
*p*-value	0.007	0.561	0.661	0.367	0.077	0.000	0.837

Kruskal–Wallis H test; one-way analysis of variance; HBMS-CCPST, Health Belief Model Scale for Cervical Cancer and Pap Smear Test.

**Table 6 healthcare-13-00804-t006:** Health Belief Model Scale results for the participants by sexual health variables.

Variables	Total Score	HBMS-CCPST	Perceived Susceptibility	Perceived Severity	Perceived Benefits	Health Motivation	Perceived Barriers
Contraceptive use status							
Yes	7 (1–17)	104 (61–137)	8 (3–13)	22 (9–35)	32 (8–40)	9.5 (4–15)	33 (14–58)
No	7 (0–17)	103 (70–136)	8 (3–13)	23 (9–33)	32 (8–40)	9 (3–15)	32 (14–58)
Total	7 (0–17)	103.5 (61–137)	8 (3–13)	22 (9–35)	32 (8–40)	9 (3–15)	32 (14–58)
*p*-value	0.423	0.342	0.478	0.652	0.317	0.268	0.965
STD risk perception							
Yes	7 (2–17)	107 (61–130)	8 (6–12)	23 (10–28)	32 (8–40)	10 (5–15)	34 (22–48)
No	7 (0–17)	102 (65–137)	7 (3–13)	22 (9–35)	32 (9–40)	9 (4–15)	32 (14–58)
No idea	6 (0–16)	111 (70–136)	8 (3–13)	24 (9–33)	32 (8–40)	9 (3–15)	36 (14–58)
Total	7 (0–17)	103.5 (61–137)	8 (3–13)	22 (9–35)	32 (8–40)	9 (3–15)	32 (14–58)
*p*-value	0.062	0.001	0.022	0.049	0.366	0.952	0.000
Need for screening							
Yes	8 (0–17)	104.53 ± 11.92	8 (3–13)	23 (9–33)	32 (8–40)	10 (3–15)	33 (14–58)
No	6 (0–16)	102.283 ± 11.023	6 (3–13)	21 (9–34)	32 (9–40)	9 (4–15)	32.5 (14–51)
No idea	6 (1–12)	105.067 ± 11.176	8 (3–12)	24 (12–35)	32 (13–40)	9 (6–13)	32 (18–55)
Total	7 (0–17)	103.992 ± 11.645	8 (3–13)	22 (9–35)	32 (8–40)	9 (3–15)	32 (14–58)
*p*-value	0.000	0.256	0.001	0.140	0.821	0.791	0.896
Perceived difficulty in accessing information about recommended screening procedures							
Very difficult/difficult	5.5 (0–13)	107 (80–130)	8 (5–13)	23 (9–33)	32 (8–40)	9 (3–12)	36.5 (14–58)
Easy	7 (1–17)	103 (61–136)	8 (3–13)	22 (9–35)	32 (8–40)	10 (4–15)	32 (14–56)
Very easy	8.5 (2–17)	100.5 (65–137)	8 (3–13)	20 (9–32)	32 (9–40)	10 (3–15)	32 (14–58)
No idea	4 (0–12)	108 (88–126)	8 (3–12)	24 (14–32)	31 (16–40)	9 (4–15)	36 (27–54)
Total	7 (0–17)	103.5 (61–137)	8 (3–13)	22 (9–35)	32 (8–40)	9 (3–15)	32 (14–58)
*p*-value	0.000	0.002	0.892	0.055	0.117	0.032	0.001

Kruskal–Wallis H test; one-way analysis of variance; HBMS-CCPST, Health Belief Model Scale for Cervical Cancer and Pap Smear Test; STD, sexually transmitted disease.

## Data Availability

The data presented in this study are available upon request from the corresponding author due to privacy restrictions.

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
