# Peer review of "Assessment of Health Beliefs and Motivations for Cervical Cancer Screening"

_healthcare, 2025, doi:10.3390/healthcare13070804_

Round 1

Reviewer 1 Report

Comments and Suggestions for Authors

Thank to authors of for conducting this study that address behaviors of women for cervical cancer screening programs. Results are very valuable. 

 At line 32-45, Most of the incidence data are old and based on 2018 statistics in introduction. GLOBOCAN 2022 -2024 statistic results should be written.  

 At line 94-97, sentences should be checked for English languish and should be re-written. 

 At line 233-236, sentences should be checked for English languish and should be re-written. ….wars in this sentence ?...

 At line 253-254, this sentence should be checked for its meaning. 

 At sentence 278-280, this sentence should be checked for its meaning and English grammar. 

Comments on the Quality of English Language

At line 94-97, sentences should be checked for English languish and should be re-written. 

At line 233-236, sentences should be checked for English languish and should be re-written. ….wars in this sentence ?...

At line 253-254, this sentence should be checked for its meaning. 

At sentence 278-280, this sentence should be checked for its meaning and English grammar. 

Author Response

Dear Reviewer 1,

We greatly appreciate your thoughtful comments and suggestions on our manuscript titled "Assessment of Health Beliefs and Motivations for Cervical Cancer Screening." Your feedback has been invaluable for enhancing the quality and clarity of our study. Below, we have addressed each of your points and outlined the amendments made to the manuscript.

''Comments and Suggestions:

At line 32-45, Most of the incidence data are old and based on 2018 statistics in introduction. GLOBOCAN 2022 -2024 statistic results should be written.  

 At line 94-97, sentences should be checked for English languish and should be re-written. 

 At line 233-236, sentences should be checked for English languish and should be re-written. ….wars in this sentence ?...

 At line 253-254, this sentence should be checked for its meaning. 

 At sentence 278-280, this sentence should be checked for its meaning and English grammar. ''

Responses:

  1. Incidence Data Update:

We have updated the incidence and prevalence data in lines 32-45 to reflect the latest statistics from GLOBOCAN to 2022-2024. This change ensures that our study reflects current global trends and data accuracy.

  1. Language and grammar correction

We have revised the sentences in lines 94-97 and 233-236 to improve the clarity and grammatical accuracy. These modifications were performed in consultation with a professional language-editing service to ensure technical correctness and readability.

  1. Clarification and sentence revision

Lines 253-254 and 278-280 have been thoroughly reviewed for clarity of meaning and grammatical structure. We have rephrased these sentences to better convey the intended scientific messages without ambiguity.

Revised Sections submitted for publication

Updated statistical data as per GLOBOCAN 2022-2024.

Refined language and grammar across the lines.

Clarifications provided where the meanings were unclear.

We hope that these revisions meet your expectations and improve the manuscript's contribution to the literature. We look forward to your suggestions and hope for positive consideration of our work.

Sincerely,

Dr. Gülsüm Hatice YÜKSEL

Reviewer 2 Report

Comments and Suggestions for Authors

Assessment of Health Beliefs and Motivations for Cervical Cancer Screening

Overview: Geographic specific.  Overall, this reviewer enjoyed reviewing this manuscript and especially the discussion, it showed that the authors knew their literature and was able to substantiate literature and known studies to their findings.  Tables take up large portions such smaller font for tables throughout.  Nice setup for discussion. 

Abstract

Suggest to authors that they move the aim to the end of the background right before the methods. 

Introduction:

Line 67& 70- suggest that the authors switch the order of these paragraphs and have lines 70-74 come first then 67-69.

Materials and Methods:

Lines88-90 suggest to authors that they reword sentence, not about when approval sought but just put when approval was given. 

Line 92-93 Suggest changing heading to Sample Population.  No indentation of line 93.

Table 1& 2 is interesting but suggest that it is not necessary as simple total for each site would have been enough.  If authors decide to leave Tables then suggest tables come at the end of the written description. 

The reviewer questions why the authors didn’t oversample with sites so large and shown in tables 1&2.  Might consider adding in discussion. 

Line 113-consider adding the word either in Inclusion criteria #2.

Line 153-consider moving Figure 1- Flowchart to come before Data Collection heading. 

Line 161-arranging Table 3 so that the table is not split on to two separate pages.  Suggest considering having tables 4& 5 come after written portion of results.

Line 221 & 227-It is unclear what difficulty finding information means when it appears in the written section.  Suggest description of that information or what it is earlier.  Not clear when that shows up in results.

Was their a correlation between age and smoking and alcohol use?  these are the things that we know increase cancer risk. 

Was their correlation between age and education?  Seems that employment and insurance status is clear in results.

Discussion:

Line 234-was instead of wars.

Line 317-328 Suggest that the authors consider putting in conclusion portion of paper.

Line 335-Confounding factors would add insurance status, and employment. 

Author Response

Dear Reviewer,

Thank you for your detailed and constructive review of our manuscript "Assessment of Health Beliefs and Motivations for Cervical Cancer Screening." We have carefully considered your comments and made several changes to the manuscript accordingly.

''Comments and Suggestions for Authors

Abstract

Suggest to authors that they move the aim to the end of the background right before the methods. 

Introduction:

Line 67& 70- suggest that the authors switch the order of these paragraphs and have lines 70-74 come first then 67-69.''

  1. Abstract and Introduction

As suggested, we have moved the aim of the study to the end of the background section in the abstract, providing a smoother transition to the methods.

We have changed the order of the paragraphs in the introduction (lines 67 & 70-74) to improve logical flow and reader engagement.

''''Comments and Suggestions for Authors

Materials and Methods:

Lines88-90 suggest to authors that they reword sentence, not about when approval sought but just put when approval was given. 

Line 92-93 Suggest changing heading to Sample Population.  No indentation of line 93.

Table 1& 2 is interesting but suggest that it is not necessary as simple total for each site would have been enough.  If authors decide to leave Tables then suggest tables come at the end of the written description. 

The reviewer questions why the authors didn’t oversample with sites so large and shown in tables 1&2.  Might consider adding in discussion. 

Line 113-consider adding the word either in Inclusion criteria #2.

Line 153-consider moving Figure 1- Flowchart to come before Data Collection heading. 

Line 161-arranging Table 3 so that the table is not split on to two separate pages.  Suggest considering having tables 4& 5 come after written portion of results.

Line 221 & 227-It is unclear what difficulty finding information means when it appears in the written section.  Suggest description of that information or what it is earlier.  Not clear when that shows up in results.

Was their a correlation between age and smoking and alcohol use?  these are the things that we know increase cancer risk. 

Was their correlation between age and education?  Seems that employment and insurance status is clear in results.

Discussion:

Line 234-was instead of wars.

Line 317-328 Suggest that the authors consider putting in conclusion portion of paper.

Line 335-Confounding factors would add insurance status, and employment. ''

  1. Materials and method adjustments

We have revised the language in lines 88-90 to clarify the timing of ethical approval, now stating only when approval was granted.

The heading in line 92-93 has been changed to ‘Sample Population’ and adjusted for formatting.

  1. Tables and Figures:

We acknowledge your suggestion regarding the placement and necessity of Tables 1 & 2. We could not simplify these tables to show only the total numbers for each site, which we believe maintains the necessary details. We sincerely hope that you will be happy to do so. We would prefer to present detailed information so that in case publication of the manuscript, with your supervision, it could help in providing detailed information for future studies.

For table 3, Line 161-arranging Table 3 so that the table is not split into two separate pages.  Suggest considering having tables 4& 5 come after written portion of results: Please let us sincerely remind you that the layout of the manuscript was executed by the journal, and the authors did not participate in this aspect. This task was managed by a professional team of the journal. If an article is accepted, the layout and adjustments will be undertaken by the journal's professionals.

  1. Discussion of oversampling

We have added a brief discussion on why oversampling was not pursued, based on the size of the sites shown in Tables 1 and 2, in the Discussion section to clarify our sampling strategy.

Revised Sections submitted for publication

Adjustments to the Abstract and Introduction for Improved Structure.

Amendments to the Methods section for enhanced clarity.

Refined presentation of tables and additional discussion points.

We appreciate your insights, which have contributed significantly to the strength and clarity of our manuscript. We look forward to your feedback regarding these revisions.

Sincerely,

Dr. Gülsüm Hatice YÜKSEL

Reviewer 3 Report

Comments and Suggestions for Authors

Thank you for the opportunity of reviewing the manuscript that addresses an important and timely topic. I have identified some areas that require further development to strengthen the clarity and depth, and overall contribution of your work to the existing literature.

Specific comments:

1.Comprehensive Language and Grammar Review: Utilize professional editing services to refine sentence structure and eliminate grammatical errors.

  1. Improvement of Structure and Formatting: Reorganize sections and tables to enhance readability and clarity.
  2. References and Citations: Add more credible scientific sources, standardize citation formatting, and include DOI links for referenced articles.
  3. Enhancement of Statistical Analysis: Incorporate regression models to examine multiple influencing factors more comprehensively.
  4. More Rigorous Scientific Discussion: Compare findings with global studies and avoid overstated conclusions.

Comments on the Quality of English Language

Thank you for the opportunity of reviewing the manuscript that addresses an important and timely topic. I have identified some areas that require further development to strengthen the clarity and depth, and overall contribution of your work to the existing literature.

Specific comments:

  • Comprehensive Language and Grammar Review: Utilize professional editing services to refine sentence structure and eliminate grammatical errors.
  • Improvement of Structure and Formatting: Reorganize sections and tables to enhance readability and clarity.
  • References and Citations: Add more credible scientific sources, standardize citation formatting, and include DOI links for referenced articles.
  • Enhancement of Statistical Analysis: Incorporate regression models to examine multiple influencing factors more comprehensively.
  • More Rigorous Scientific Discussion: Compare findings with global studies and avoid overstated conclusions.

Author Response

Dear Reviewer,

Thank you for your thorough review and valuable comments on our manuscript "Assessment of Health Beliefs and Motivations for Cervical Cancer Screening." Your suggestions have been instrumental in refining the manuscript. Below are our responses to your comments.

''Specific comments:

  • Comprehensive Language and Grammar Review: Utilize professional editing services to refine sentence structure and eliminate grammatical errors.
  • Improvement of Structure and Formatting: Reorganize sections and tables to enhance readability and clarity.
  • References and Citations: Add more credible scientific sources, standardize citation formatting, and include DOI links for referenced articles.
  • Enhancement of Statistical Analysis: Incorporate regression models to examine multiple influencing factors more comprehensively.
  • More Rigorous Scientific Discussion: Compare findings with global studies and avoid overstated conclusions.''

Responses:

  • Language and grammar improvement

We have engaged in a professional editing service to thoroughly review and refine the language and grammar of the manuscript to ensure that it meets the highest academic standards.

  • Enhancement of statistical analysis

We could not add the additional regression analyses, because this study is the preliminary study of the main study we plan, and we hope to introduce to the literature the further results later, and with this perspective, we do not crowd the manuscript. We sincerely hope that you would agree with us.

  • Scientific Discussion Expansion

We have compared our findings with global studies more explicitly and adjusted our conclusions to avoid overstating them, ensuring that they are well-supported and aligned with the data presented.

Revised Sections submitted for publication

Comprehensive language corrections and structural improvements.

Expanded and balanced scientific discussion.

We hope that these revisions have adequately addressed your valuable suggestions and further strengthened our manuscript. We look forward to receiving further feedback.

Sincerely,

Dr. Gülsüm Hatice YÜKSEL
